# Evaluation of serum uric acid and liver function tests among pregnant women with and without preeclampsia at the University of Gondar Comprehensive Specialized Hospital, Northwest Ethiopia

**Fethya Seid Hassen, Tabarak Malik⬥, Tadesse Asmamaw Dejenie⬥***

Department of Biochemistry, School of Medicine, College of Medicine and Health Sciences, University of Gondar, Gondar, Ethiopia

* as24tadesse@gmail.com

## Abstract

### Background

Pre-eclampsia can be described as new-onset hypertension (blood pressure $\geq$140/90 mmHg) together with proteinuria (24-hr urinary protein $\geq$ 0.3 g) or any indication of end-organ damage after 20 weeks of gestation. Liver and kidney dysfunction, thrombocytopenia, pulmonary edema, and neurologic dysfunction are common manifestations of end-organ damage due to pre-eclampsia. Pre-eclampsia is the most common cause of liver and kidney dysfunction due to hypoxia and endothelial dysfunction. Hyperuricemia indicates kidney dysfunction and is considered a predictor of the severity of preeclampsia. Therefore, the objective of this study is to evaluate the utility of the levels of serum uric acid and liver function tests [alanine aminotransferase (ALT) and aspartate aminotransferase (AST)] as biomarkers of preeclampsia-related organ damage.

### Methods and materials

An institutional-based comparative cross-sectional study design was conducted, and a total of 102 subjects (51 patients with preeclampsia and 51 normotensive pregnant women) were recruited. The parameters measured were levels of serum uric acid and liver function tests.

### Results and discussion

There were statistically significant differences in the mean serum uric acid, ALT, and AST levels between preeclamptic pregnant women and normotensive pregnant women (p<0.05). There were no statistically significant differences in the mean total and direct bilirubin levels. There was also a significant difference in mean serum uric acid, alanine transaminase, and aspartate transaminase levels across different gestational age categories.

**Data Availability Statement:** The authors confirm that all data underlying the findings are fully

available without restriction. All relevant data are within the manuscript.

**Funding:** The authors received no specific funding for this work.

**Competing interests:** The authors have declared that no competing interests exist.

**Abbreviations:** ALP, Alkaline Phosphatase; ALT, Alanine Transaminase; ANOVA, Analysis of Variance; AST, Aspartate Transaminase; BMI, Body Mass Index; CI, Confidence interval; DBP, Diastolic Blood Pressure; HELLP, Haemolysis Elevated Liver Enzymes and Low platelets; LDH, Lactate Dehydrogenase; LFT, Liver Function Test; SBP, Systolic Blood Pressure; SD, Standard Deviation; SST, Serum Separator Tube; SUA, Serum Uric Acid; UoG, University of Gondar; XDH, Xanthine Dehydrogenase; XO, Xanthine Oxidase.

## Conclusion

Our study revealed that serum uric acid, ALT, and AST levels were higher in pre-eclamptic pregnant women compared to those of normotensive pregnant women, and the differences were statistically significant. As such, serum uric acid and liver function tests may be considered biomarkers of pre-eclampsia-related end-organ damage.

## Introduction

Pregnancy-induced hypertension is a major cause of maternal and fetal morbidity and mortality. During pregnancy, four classes of hypertensive disorders exist. This includes chronic hypertension, gestational hypertension, preeclampsia-eclampsia, and chronic hypertension with superimposed preeclampsia [1].

Gestational hypertension develops after 20 weeks of gestation and has no diagnostic criteria for preeclampsia, and it can change to chronic hypertension if the blood pressure remains increased more than 12 weeks postpartum. Almost 15–46% of women will develop preeclampsia among those initially diagnosed with gestational hypertension [2]. This represents the highest degree of feto-maternal complications [3].

Pre-eclampsia is defined as new-onset hypertension (blood pressure ≥140/90 mmHg) in combination with proteinuria (24-hr urinary protein ≥ 0.3 g) or any sign of end-organ damage after 20 weeks of gestation. The presence of 300 mg or more of protein in a 24-hour urine collection or a urine dipstick protein of +1 is termed proteinuria [4]. Proteinuria, however, is not a requirement anymore to make a diagnosis of pre-eclampsia [5].

Disease severity varies, and it is considered severe if it is associated with one or more of the following; elevated blood pressure ≥160/110 mmHg on two circumstances at least 6 hours apart, a 24-hour protein of ≥5g, and/or other symptoms, such as headache, blurring of vision, vomiting. The main underlying disease mechanism is endothelial dysfunction affecting multiple organs, including the brain, liver, kidneys, and placenta [5, 6] in conjunction with other symptoms such as impaired liver and kidney function, oliguria, headache, hyperreflexia, or right upper quadrant and epigastric pain, and thrombocytopenia (hemolysis elevated liver enzymes and low platelets syndrome (HELLP)) [6, 7].

Approximately 50,000 maternal deaths worldwide are caused annually by preeclampsia, of which 25% are due to intrauterine growth restrictions (IUGR) and 15% are due to preterm birth in developed countries [8].

Its global prevalence is extremely variable and ranges from 2–10% of pregnancies. The World Health Organization reports that the incidence in developing countries (2.8% of live births) is sevenfold greater than that in developed countries (0.4%) [9]. In Africa, it occurs in 10% of pregnancies, which is higher than the global average. Around 4 in 100 women experience problems with high blood pressure and kidney dysfunction [10]. Ethiopian National Emergency Obstetric and Newborn Care found that approximately 5 percent of all pregnancies and 1 percent of all deliveries result in complications from preeclampsia. The UN 2010 study found that Ethiopia is one of the five countries that accounts for 50 percent of the world's maternal deaths, most of which are due to preeclampsia [11].

Liver and kidney dysfunctions are common manifestations of end-organ damage due to preeclampsia [4].

Hyperuricemia indicates kidney dysfunction because of decreased glomerular filtration, decreased tubular secretion, and/or increased proximal tubular reabsorption [12]. It is also an

independent risk factor for cardiovascular diseases as it has been suggested that it alters vascular function and mediates vascular inflammation. As such, hyperuricemia can perhaps predict the severity of pre-eclampsia [13].

Pre-eclampsia is also the most common cause of liver dysfunction in 3% of pregnancies because of microvesicular fat deposition and reduced blood flow to the liver potentially causing ischemia and periportal hemorrhage [14, 15]. Alanine aminotransferase (ALT) and aspartate aminotransferase (AST) levels are usually normal, but when they become elevated and are accompanied by abdominal pain, it almost always suggests the severe end of the disease spectrum [16–18].

As such, this study was undertaken to evaluate the utility of levels of serum uric acid and liver function tests as biomarkers of pre-eclampsia-related end-organ damage among pregnant women with and without preeclampsia.

## Methods and materials

### Study design and patients

An institutional-based comparative cross-sectional study design was implemented. We enrolled 102 participants (51 preeclamptic pregnant women as confirmed by physicians and 51 normotensive pregnant women). All patients had antenatal care follow-up at the University of Gondar Comprehensive Specialized Hospital. We excluded pregnant women who had chronic hypertension, liver disease, kidney disease, gout, cardiac disease, diabetes mellitus, infections, a history of medication use (e.g., aspirin, phenytoin, tetracycline, sulphonamides, etc.), and those with substance abuse (eg smoking and alcohol consumption) as these factors may affect the outcome variables.

### Study area and period

The study was conducted from February to October 2020 at the University of Gondar Comprehensive Specialized Hospital, Northwest Ethiopia, to evaluate the levels of serum uric acid and liver function tests among pregnant women with and without preeclampsia.

### Sample size determination

To determine the sample size, G-Power Version 3.1.9.2 was used as a tool. G-Power is one of the software packages that perform sample size calculations such as Minitab and Epi-info, covering a wider range of study designs. As an input, G-Power requires selecting an appropriate test family (t-test in this case) and statistical test within the test family (linear regression in this case) and specifying alpha error probability, power (1-β error probability), and effect size. By considering alpha = 0.05, power (1- β) = 0.8 (80%), and effect size = 0.5, the total sample size was 102.

The number of participants enrolled in the study was 102 (51 preeclamptic pregnant women as confirmed by physicians and 51 normotensive pregnant women).

### Data collection methods, blood sample collection, and processing

Trained nurses collected sociodemographic data after careful examination of the patients' histories. Four laboratory technicians were oriented on blood sample collection and storage. Questionnaires were filled out through face-to-face interviews with participants. Pre-pregnancy BMIs were calculated from prepregnancy body weight (kg) and height (meter) as follows: BMI = Weight (in kg)/(Height in m$^{)}$ $^2$. Using the WHO (2008), classification of five categories of BMI can be identified as follows: underweight, <18.5 kg/m$^2$; normal, (18.5–24.9

$kg/m^{2)}$); overweight, (25.0–29.9 $kg/m^{2)}$); and obesity, 30.0–34.9 or 35.0–39.9 $kg/m^2$, extreme obesity > 40.0 $kg/m^2$ [19]. Two consecutive blood pressure readings were also measured.

Approximately 5 ml of venous blood was collected in a sterile plain vacutainer tube and properly labeled with a specific code of the patient. Blood collected in SST tubes was allowed to stand for 30 minutes at room temperature to allow complete clotting and clot retraction. It was then centrifuged (MEGAFUGE R 1.0 HERAEUS) at 3500 rpm for 15–30 min to extract serum. The serum was kept at -80˚C refrigerators until biochemical analysis was carried out. The extracted serum was used to determine the biochemical levels (uric acid, aspartate aminotransferase (AST), and alanine aminotransferase (ALT)) and analyzed using Biosystem kits on a Mindray BS200E Chemistry Analyser at the UoG main lab.

## Data quality control and management

To assure data quality, orientation was given to data collectors on the objective of the study, data collection process, and relevance of the study before data collection. The validity of the questionnaire was maintained by a pretested questionnaire. Quantitative data were double entered to check if there is any data inconsistency and to avoid any problem with data entry processes. The data collectors were supervised daily throughout the data collection. The data were checked for completeness on-site, and before data entry, the incomplete data were discarded. All the laboratory procedures were handled by professional laboratory technicians. All the laboratory instruments were automated and standardized.

## Data processing and analysis

All data were checked for cleaning and completeness. Data processing and analysis obtained from laboratory analyses of the blood samples and questionnaires were performed by coding and entering the data into Epi-Data version 4.6.0 statistical software. The data were exported to STATA 14.0 software for analysis. Continuous variables are expressed as the mean ± standard deviation (SD), while categorical variables are presented as frequencies and percentages. Tables and graphs were also used to present the data. Levene's test of homogeneity of variances, Shapiro–Wilk test of normality, multicollinearity diagnosis, and other assumptions was checked before performing any statistical analysis and were fulfilled. Descriptive analyses, independent sample t-tests, and ANOVA were performed. Bivariable and multivariable linear regression analyses were used to assess the association of different variables. With bivariable linear regression analysis, variables with p values <0.2 were selected for multivariable analysis while with multivariable linear regression analysis, variables with p values<0.05 at 95% confidence intervals were considered statistically significant in all analyses.

## Ethics approval and consent to participate

Ethical clearance was obtained from the University of Gondar College of Medicine and Health Sciences School of Medicine with protocol number 1995/05/20. The objectives of the study were briefly explained to the participants in a local language (Amharic), and clarification was given. Verbal permission and written consent were also obtained from each study participant before starting data collection.

## Results

### Sociodemographic characteristics of study participants

The present study recruited 102 female study participants, 51 of whom were pre-eclamptic pregnant women while 51 of who were normotensive pregnant women with gestational age

above 20 weeks. The mean ages of preeclamptic patients and controls were 32.9 and 29.5 years with a minimum age of 21 and 23 years and a maximum age of 42 and 35 years, respectively. The mean gestational ages of pre-eclamptic patients and normotensive pregnant women were 33.6 weeks (range: 22–39) and 29.3 weeks (range: 21–38) respectively. The majority (98%) of the study participants were married. Among the preeclamptic patients, 31 (60.8%) were living in rural areas. Among the normotensive pregnant women, 41 (80.4%) were living in urban areas. Most of the 44 (86.3%) and 41 (80.4%) patients were Orthodox Christians, and the rest were Muslims in the case and control groups, respectively. Most of the study participants were housewives (76.5%) (Table 1).

## Anthropometric and clinical characteristics of study participants

Prepregnancy BMI was calculated using their prepregnancy weight and height. Based on the calculation, the majority of the patients (41, 80.4%) had a normal BMI, 8 (15.7%) were overweight, and 2 (3.9%) were obese. The majority of the normotensive pregnant women (94.1%) had normal BMI while the rest (5.9%) were overweight. None in the control group was underweight. Blood pressures of the study participants were recorded and found to be high in preeclampsia patients, with a mean SBP of 142.8±6.34 and a mean DBP of 92.8±5.22.

## Descriptive characteristics of serum uric acid and liver function tests between preeclampsia patients and normotensive pregnancy

The normal concentrations of serum uric acid for females were 2.4–5.7 mg/dl, serum ALT 0–31 IU/L, serum AST 0–32 IU/L, serum total bilirubin 0–1.2 mg/dl, and serum direct bilirubin 0–0.25 mg/dl. The majority of the preeclamptic patients (70.6%, 80.4%, and 90.2%) had above normal serum uric acid (SUA), ALT, and AST levels, respectively but had normal total and direct bilirubin levels (60.8% and 62.7% respectively). Most of the normotensive pregnant women had normal SUA, ALT, AST, and total and direct bilirubin levels.

An independent sample t-test showed that there was a statistically significant difference in the mean serum uric acid level between preeclamptic pregnant women and normotensive pregnant women (p-value = 0.001). Similarly, there was also a statistically significant difference in the mean serum ALT (p = 0.002) and mean serum AST levels (p = 0.001) between preeclamptic women and normotensive pregnant women. There was, however, no significant difference in the total and direct bilirubin levels in preeclamptic pregnant women were not significantly different from those in normotensive pregnant women (p = 0.135 and p = 0.181, respectively). The age and gestational age of the study participants were also not significantly different between the case and control groups (p = 0.059 and p<0.061). Pre-pregnancy BMI was not significantly higher in preeclamptic patients compared to that of normotensive pregnant women (p = 0.580) (Table 2).

## Comparison of mean SUA, ALT, and AST levels across different age groups of study participants

A post hoc (Bonferroni) test showed that there was a significant difference in mean serum ALT between the age group of 26–34 years and age $\geq$ 35 years (p = 0.005). However, there was no significant difference in the mean ALT levels between 19-25-year-old and 26-34-year-old (p = 0.72). There was also no significant mean ALT difference between the age groups 19–25 years and $\geq$ 35 (p = 0.19).

A post hoc (Bonferroni) test also showed that there was a significant difference in mean AST level between the age group of 26–34 years and age $\geq$ 35 years (p = 0.006). There was also

**Table 1. Baseline characteristics of study participants, University of Gondar Comprehensive Specialized Hospital, Northwest Ethiopia, 2020.**

| Variables | Preeclampsia (51) | | Normotensive pregnancy (51) | | |
|---|---|---|---|---|---|
| | Frequency (count) | % | Frequency(count) | % | P-value |
| **Age (years)** | | | | | |
| 19–25 | 11 | 21.7 | 18 | 35.3 | 0.851 |
| 26–34 | 12 | 23.5 | 24 | 47.06 | |
| ≥35 | 28 | 54.9 | 9 | 17.7 | |
| **Marital status** | | | | | |
| Single | 1 | 1.96 | 1 | 1.96 | 0.095 |
| Married | 50 | 98.1 | 50 | 98.04 | |
| **Residence** | | | | | |
| Urban | 20 | 39.2 | 41 | 80.4 | 0.302 |
| Rural | 31 | 60.8 | 10 | 19.6 | |
| **Educational status** | | | | | |
| Illiterate | 21 | 41.2 | 11 | 21.6 | |
| Only read and write | 9 | 17.6 | 5 | 9.8 | 0.41 |
| Elementary | 14 | 27.5 | 27 | 52.9 | |
| school | 5 | 9.8 | 7 | 13.7 | |
| High school College and above | 2 | 3.9 | 1 | 1.96 | |
| **Occupational status** | | | | | |
| Housewife | 39 | 76.5 | 39 | 76.5 | 0.235 |
| Gov. employee | 2 | .92 | 1 | 1.96 | |
| Self-employed | 10 | 19.6 | 10 | 19.6 | |
| Unemployed | 0 | 0 | 1 | 1.96 | |
| **Gestational age at diagnosis (in weeks)** | | | | | |
| 20–30 | 10 | 19.6 | 28 | 54.9 | 0.061 |
| 31–35 | 12 | 23.5 | 13 | 25.5 | |
| ≥36 | 29 | 56.7 | 10 | 19.6 | |
| **Previous history of preeclampsia** | | | | | |
| Yes | 8 | 15.7 | 4 | 7.8 | 0.062 |
| No | 43 | 84.3 | 47 | 92.2 | |
| **Family history of preeclampsia** | | | | | |
| Yes | 1 | 1.196 | 1 | 1.96 | 0.04 |
| No | 50 | 98.04 | 50 | 98.04 | |
| **Parity** | | | | | |
| Nullipara | 31 | 60.8 | 4 | 11.8 | 0.213 |
| ≥1 parity | 20 | 39.2 | 47 | 88.2 | |
| **Fetus of the current pregnancy** | | | | | |
| Singleton | 41 | 80.4 | 49 | 96.1 | 0.14 |
| Twin | 10 | 19.6 | 2 | 3.9 | |
| **BMI** | | | | | |
| <18.5Kg/m$^2$ | 0 | 0 | 0 | 0 | 0.714 |
| 18.5–24.9 Kg/m$^2$ | 41 | 80.4 | 48 | 94.1 | |
| 25–29.9 kg/m$^2$ | 8 | 15.7 | 3 | 5.9 | |
| ≥30 kg/m$^2$ | 2 | 3.9 | 0 | 0 | |
| **SBP** | | | | | |
| ≤ 140 mmHg | 14 | 27.4 | 1 | 1.96 | |
| ≥ 140 mmHg | 37 | 72.6 | 50 | 98.04 | |
| **DBP** | | | | | |

*(Continued)*

**Table 1.** (Continued)

| Variables | Preeclampsia (51) | | Normotensive pregnancy (51) | | |
|---|---|---|---|---|---|
| | Frequency (count) | % | Frequency(count) | % | P-value |
| ≤ 90 mmHg | 14 | 27.4 | 1 | 1.96 | |
| ≥90 mmHg | 37 | 72.6 | 50 | 98.04 | |

a significant difference in mean AST level between the age group 19–25 years and age ≥ 35 years (p = 0.025). However, there was no significant mean AST level difference between the 19-25-year-old and 26-34-year-old age groups (p = 1.000).

## Comparison of mean SUA, ALT, and AST levels at different gestational age categories of study participants

The post hoc (Bonferroni) test showed that there was a significant difference in mean ALT between the gestational age groups of 20–30 weeks and ≥ 36 weeks (p<0.001). However, there was no significant mean ALT level difference between the gestational ages of 20–30 weeks and 31–35 weeks (p = 0.23). Similarly, there was also no significant difference in mean ALT between the age groups 31–35 weeks and ≥ 36 weeks (p = 0.255).

A post hoc (Bonferroni) test also showed that there was a significant difference in mean AST level between the gestational age groups of 20–30 weeks and ≥ 36 weeks (p<0.001). However, there was no significant mean AST level difference between the gestational ages of 20–30 weeks and 31–35 weeks (p = 0.144). Similarly, there was also no significant difference in mean AST between the 31-35-week and ≥ 36-week gestational age groups (p = 0.344).

## Multiple linear regression analysis of SUA with different predictor variables of study participants

Variables with a p-value less than 0.2 in the bivariable linear regression analysis were selected for multivariable linear regression analysis. Bivariable linear regression analysis of all predictor variables with SUA level was performed separately, and only age, gestational age, BMI, systolic blood pressure, diastolic blood pressure, and preeclamptic status had a p-value of less than 0.2. The remaining variables were not eligible for multiple linear regressions.

**Table 2. Independent t-test for mean comparison of serum uric acid and liver function tests between preeclamptic and normotensive pregnant women, University of Gondar Comprehensive Specialized Hospital, Northwest Ethiopia.**

| | Preeclampsia (n = 51) | Normotensive pregnancy (n = 51) | |
|---|---|---|---|
| | Mean (95% CI) | Mean (95% CI) | p value |
| SUA (mg/dl) | 6.17 ±1.04 | 3.65± 1.19 | 0.001* |
| ALT(IU/L) | 35.9± 7.28 | 20.2 ± 7.23 | 0.002* |
| AST(IU/L) | 39.7± 6.73 | 20.5±7.65 | 0.001* |
| Bilirubin total(mg/dl) | 0.95± 0.45 | 0.8±0.27 | 0.135 |
| Bilirubin direct(mg/dl) | 0.31±0.12 | 0.276±0.2 | 0.181 |
| Age (years) | 32.9±6.3 | 29.5±3.3 | 0.059 |
| Gestational age (in Weeks) | 33 .5±4.3 | 29.4±4.9 | 0.061 |
| Prepregnancy BMI | 23.4±0.51 | 21.5±0.23 | 0.580 |

*p<0.05 is statistically significant

In the multivariable linear regression analysis, age, SBP, DBP, gestational age at diagnosis, and being preeclamptic were significantly associated with SUA level (p = 0.041, p = 0.013, p = 0.041, p = 0.034 p = 0.01, respectively). BMI (P = 0.191) did not show a significant association with SUA. Keeping other variables constant, a one-year increase in age increases SUA by a factor of 0.81, a one-unit increase in SBP increases SUA by a factor of 0.4 and a one-unit increase in DBP increases SUA level by a factor of 0.2. A one-week change in gestational age increased/decreased SUA level by a factor of 0.18. Being preeclamptic increased SUA levels by a factor of 2.37.

## Multiple linear regression analysis of ALT with different predictor variables of study participants

Bivariable linear regression analysis of all predictor variables with ALT was performed separately, and only age, gestational age at diagnosis, prepregnancy BMI, systolic blood pressure, diastolic blood pressure, parity, and preeclamptic status had a p-value less than 0.2. The remaining variables were not eligible for multiple linear regressions.

In the multivariable linear regression analysis, age, SBP parity, and preeclamptic status were significantly associated with ALT (p = 0.045, p = 0.046, p = 0.016, p = 0.001, respectively). Keeping other variables constant, a one-year increase in age increases the level of ALT by a factor of 0.51, a one-unit increase in SBP of respondents increases ALT by a factor of 0.2, and being nullipara increases ALT by a factor of 1.2. Being preeclamptic increases serum ALT by a factor of 2 (Table 3).

## Multiple linear regression analysis of AST level with different predictor variables of study participants

Bivariable linear regression analysis of all predictor variables with AST was performed separately, and age, gestational age at diagnosis, prepregnancy BMI, systolic blood pressure, diastolic blood pressure, parity, and preeclampsia status had a p-value <0.2.

In the multivariable linear regression analysis, age, SBP, DBP, and being preeclamptic were significantly associated with AST level (p = 0.031, p = 0.002, p = 0.037, p = 0.015, respectively). Keeping other variables constant, a one-year increase in age increases AST level by a factor of 1.56, a one-unit increase in SBP increases AST level by a factor of 0.42 and a one-unit increase

**Table 3. Multiple linear regression analysis of SUA, ALT, AST, and bilirubin levels with different predictor variables of preeclamptic women, University of Gondar Comprehensive Specialized Hospital, Northwest Ethiopia, 2020.**

| Variables | SUA | ALT | AST | Total bilirubin | Direct bilirubin |
|---|---|---|---|---|---|
| | β(95%CI, p value) | β(95%CIp value) | β(9%CI, p value) | β(95%CI,p- value) | β (95%CI, p value) |
| Age | 0.81,0.04 | 0.51,0.4 | 1.56,0.031 | 0.07,0.153 | 1.9,0.89 |
| Gestational age | 0.18,0.03 | 0.88,0.3 | 1.51,0.53 | 0.052,0.89 | 0.0073,0.9 |
| SBP | 0.4,0.013 | 0.2,0.05 | 0.42,0.002 | 0.054,0.962 | 0.04,0.72 |
| DBP | 0.2,0.013 | 0.1,0.17 | 0.1,0.037 | 0.06,0.44 | 0.05,0.62 |
| BMI | 1.82,0.19 | 0.87,0.6 | 2.43,0.26 | 1.4,0.72 | 0.52,0.63 |
| Nulliparity | 0.4,0.2 | 0.12,0.02 | 0.78,0.48 | 0.3,0.07 | 0.04,0.09 |
| Being preeclamptic | 2.37,0.01 | 2.02,0.001 | 1.9,0.015 | 0.8,0.021 | 0.65,0.02 |
| Previous history of PE | 1.3,0.7 | 3.3,0.15 | 2.7,0.24 | 1.4,0.07 | 2.9,0.61 |
| Family history of PE | 0.74,0.34 | 3.9,0.09 | 1.62,0.8 | 2.3,0.87 | 0.58,0.45 |
| No of fetus of pregnancy | 0.9,0.056 | 1.7,0.14 | 0.36,0.08 | 2.7,0.65 | 0.22,0.76 |

* P values are considered significant when <0.05.

in DBP increases AST level by a factor of 0.06. Being preeclamptic increases serum AST by a factor of 1.9 (Table 3).

## Multiple linear regression analysis of total bilirubin with different predictor variables of study participants

Bivariable linear regression analysis of all predictor variables with total bilirubin was performed, and age, gestational age at diagnosis, systolic blood pressure, diastolic blood pressure, and preeclamptic status had a p-value less than 0.2. However, only preeclamptic status showed a significant association with total bilirubin in multiple linear regressions (p = 0.025). Being preeclamptic increased the total bilirubin level by a factor of 0.8 (Table 3).

## Multiple linear regression analysis of direct bilirubin with different predictor variables of study participants

Bivariable linear regression analysis of all predictor variables with direct bilirubin was performed, and gestational age at diagnosis, systolic blood pressure, diastolic blood pressure, and preeclamptic status had a p-value less than 0.2. Only being preeclamptic showed a significant association with direct bilirubin in multiple linear regressions (p = 0.02) (Table 3).

## Discussion

In our study, the majority of the pre-eclamptic patients were found ≥35 years old which is in keeping with the results of a study conducted by Tessema *et al.*, and Jasovic-Siveska, *et al.*, showing that advanced maternal age is associated with the development of preeclampsia [20, 21]. Other studies conducted in Finland and Japan also demonstrated that women of advanced maternal age were 1.5 times more likely to have preeclampsia than women under the age of 35 years. This may be related to the aging of the uterine blood vessels [22, 23].

Our study also showed that the mean ± SD of prepregnancy BMI was found to be 32.4± 0.51 kg/m$^2$ and 21.5 ± 0.23 kg/m$^2$ for the case and control groups, respectively, and there was no statistically significant difference between the two groups. This finding is in contrast with the findings of other studies [24, 25], and this difference may be due to nutritional variation, sample size variation, and/or lifestyle differences.

In our study, the mean serum uric acid level was significantly higher in cases compared to controls (p = 0.001) which is again consistent with the findings of other studies [15, 26].

In addition, the mean serum ALT and AST levels were also significantly higher. Other researchers also noticed an increase in serum ALT levels in preeclampsia [15, 17]. This elevation may be due to placental ischemia followed by a systemic inflammatory response, resulting in endothelial dysfunction causing vasoconstriction and eventual liver and kidney dysfunctions. Other similar studies conducted by Ekun et al. also concluded that there was an increase in transaminases in preeclamptic patients compared to controls [27].

On the other hand, our study showed that there was no significant difference in the total and direct bilirubin levels in preeclamptic pregnant women and those in normotensive pregnant women. Our findings are in contrast with the findings of other studies wherein there were significant increases in serum bilirubin [28], alanine transaminase, and aspartate aminotransferase levels in preeclamptic women compared to normotensive pregnant women [29, 30]. This discrepancy may be due to variations in the extent of hemolysis and/or reference value of the chemistry analyzer. In pre-eclampsia, the main cause of hyperbilirubinemia is thought to be disseminated intravascular coagulation caused by hemolysis and liver cell necrosis [16].

In our study, there were significant differences in the mean SUA, ALT, and AST levels across the different age groups (p = 0.0023, p = 0.0063, p = 0.0037, respectively). The mean SUA level in the age group 26–34 years was 4.43 mg/dL, which then increased to 5.66 mg/dL in the age group >35 years (p = 0.002). The mean ALT level in the 26-34-year-old age group was 24.41 U/L and increased to 32.18 U/L in the age group ≥35 years (p = 0.005). The mean AST level among the 26–34 years age group was 26.17 U/L and increased to 34.93 U/L in the age group ≥35 years (p = 0.006). These findings were consistent with those of other studies, which showed that serum uric acid and liver function tests increased with advancing maternal age [31, 32].

Similarly, there were also significant differences in the mean SUA, ALT, and AST levels across the different gestational age groups (p = 0.005, p = 0.007, p = 0.0005, respectively). The mean SUA levels among the gestational age group of 20–30 weeks was 4.35 mg/dl and increased to 5.69 mg/dl in the gestational age group of ≥ 36 weeks (p<0.001). The mean ALT in the gestational age group of 20–30 weeks was 23.49 U/L and increased to 32.55 U/L in the gestational age group of ≥36 weeks (p<0.001). The mean AST level among the gestational age group of 20–30 weeks was 24.02 U/L and increased to 34.75 U/L in the gestational age group of ≥36 weeks (p<0.001). This was in line with previous studies reporting that higher gestational age was associated with preeclampsia and resulted in high serum uric acid and liver function tests [15].

Lastly, we performed a multivariable linear regression analysis and found a statistically significant positive association between SUA and age, SBP, DBP, gestational age, and preeclamptic status (p = 0.041, p = 0.013, p = 0.041, p = 0.034, p = 0.01, respectively). A study conducted in Ukraine and Europe found that SUA levels increased with increasing age and gestational age[33, 34].

In the present study, preeclamptic patients were positively and significantly associated with SUA, ALT, and AST levels. Preeclampsia was associated with a marked increase in SUA, AST, and ALT levels compared to controls. This was in line with other studies [29, 30].

## Conclusion

Our study revealed that serum uric acid, ALT, and AST levels were higher in pre-eclamptic pregnant women compared to those of normotensive pregnant women, and the differences were statistically significant. The study also shows that a significant difference was observed in mean SUA, ALT, and AST levels across different age and gestational age categories among pre-eclamptic patients.

## Author Contributions

**Conceptualization:** Fethya Seid Hassen.

**Data curation:** Fethya Seid Hassen, Tabarak Malik.

**Formal analysis:** Tadesse Asmamaw Dejenie.

**Investigation:** Fethya Seid Hassen.

**Methodology:** Tabarak Malik, Tadesse Asmamaw Dejenie.

**Resources:** Tabarak Malik.

**Supervision:** Tabarak Malik.

**Validation:** Tadesse Asmamaw Dejenie.

**Visualization:** Tadesse Asmamaw Dejenie.

**Writing – original draft:** Fethya Seid Hassen.

**Writing – review & editing:** Tadesse Asmamaw Dejenie.

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
