## [Decision Letter · Decision Letter 0]

12 Apr 2022

PONE-D-21-30928EVALUATION OF SERUM URIC ACID AND LIVER FUNCTION TESTS AMONG PREGNANT WOMEN WITH AND WITHOUT PREECLAMPSIA AT UNIVERSITY OF GONDAR COMPREHENSIVE SPECIALIZED HOSPITAL, NORTHWEST ETHIOPIAPLOS ONE

Dear Dr. Dejenie,

Thank you for submitting your manuscript to PLOS ONE. After careful consideration, we feel that it has merit but does not fully meet PLOS ONE’s publication criteria as it currently stands. Therefore, we invite you to submit a revised version of the manuscript that addresses the points raised during the review process.

 Reconsider your conclusions that the biomarkers measured are associated with early stages of preeclampsia

Please address reviewers comments

We look forward to receiving your revised manuscript.

Kind regards,

Maria Lourdes Gonzalez Suarez, MD, PhD

Academic Editor

PLOS ONE

Journal Requirements:

NO

We would like to express our gratitude to …the University of Gondar 

College of Medicine and Health Sciences for the financial, material, and equipment support 

for this research work.

NO

NO

Additional Editor Comments:

Thank you for submitting your manuscript. I apologize for the delay during the peer review process. Please address the comments made by our reviewers, especially your suggestions regarding liver function panel to be considered as biomarker of early stages of preeclampsia, please provide data on when the samples where obtained regarding gestational age.

Reviewers' comments:

Reviewer's Responses to Questions

**Comments to the Author**

1. Is the manuscript technically sound, and do the data support the conclusions?

Reviewer #1: No

Reviewer #2: Yes

2. Has the statistical analysis been performed appropriately and rigorously? 

Reviewer #1: No

Reviewer #2: I Don't Know

3. Have the authors made all data underlying the findings in their manuscript fully available?

Reviewer #1: No

Reviewer #2: Yes

4. Is the manuscript presented in an intelligible fashion and written in standard English?

Reviewer #1: Yes

Reviewer #2: Yes

5. Review Comments to the Author

Reviewer #1: The manuscript by Dejenie et al. explores common biochemical biomarkers such as uremic acid and liver function enzymes, ALT and AST, in pregnant women with and without preeclampsia. While the manuscript provides valuable insight into the biochemical profile of women with preeclampsia, there is some major insufficiency.

Minor comments:

1. Table 4: it is not clear which column is "Cases" and which is "Controls"

2. t-test was applied on several spots where it was obvious from presented data that CV was above 30%

3. The authors did not address early vs. late preeclampsia differences.

4. Comparison studies of biochemical markers across gestational age and different age groups would look better it it were presented in tables rather than textually.

Major comments:

1. Based on the results we can only conclude that there are differences among PE vs normotensive pregnancies. However the authors make a conclusion of SUA, ALT and AST as early biomarkers of preeclampsia. This is incorrect, as they did not report when were the samples collected. Based on their study design, these women have already been diagnosed with PE so increase of these biochemical parameters could be the consequence of the disease, not early marker of it.

2. The authors report that women with preeclampsia were older compared to the control group. In the same time they observe positive association between age and biochemical markers such as ALT and AST. It is possible that age is actually pulling the difference so authors should have performed adequate statistical analysis to check this.

3. I am confused that the authors did not observe the difference in bilirubin concentrations between cases and controls, but report "being preeclamptic shows significant association with total bilirubin in multiple linear regressions"

Reviewer #2: Review:

In this cross-sectional study, Dr. Hassen et al aimed to evaluate the utility of levels of serum uric acid and liver function tests/transaminases as early biomarkers of pre-eclampsia-related end organ damage. Both elevated serum uric acid and liver function tests/transaminases are potential laboratory findings in pre-eclampsia. Although the findings are not necessarily novel, the text is clear and easy to read, but I do have some suggestions as outlined below. There are also inconsistencies in some of the authors’ statements, which need to be corrected/modified.

Abstract:

Line 11: Pre-eclampsia can be described as new-onset hypertension…

Line 12: …or ANY indication…

Line 13: Liver and kidney dysfunction…

Line 14: …common MANIFESTATIONS of end-organ damage…

Line 15: …most common cause of LIVER and KIDNEY dysfunction…(Please change all “renal” terms to “kidney” instead.)

Line 15: Pre-eclampsia is the most common cause of liver and kidney dysfunction – in what patient population? – due to hypoxia and endothelial dysfunction.

Line 17: Hyperuricemia indicates kidney dysfunction…

Line 18: …severity of pre-eclampsia; therefore, the objective of this study is to evaluate the utility of the levels of serum uric acid and liver function tests [alanine aminotransferase (ALT) and aspartate aminotransferase (AST)] as early biomarkers…

Line 20: An institution-based comparative cross-sectional study design…

Line 21: …conducted, …

Line 22: The parameters measured were levels of serum uric acid and liver function tests.

Line 24: There were statistically significant differences in the mean serum uric acid, ALT, and AST levels …

Line 26: There were no statistically significant differences in the mean total and direct bilirubin levels…

Line 27: There were also statistically significant differences in the mean serum uric acid, ALT, and AST levels across different gestational age categories.

Line 30: Our study revealed that serum uric acid, ALT, and AST levels were HIGHER in pre-eclamptic pregnant women compared to THOSE of healthy pregnant women, and the differences were statistically significant. As such, serum uric acid and liver function tests may be considered as early biomarkers of pre-eclampsia-related end organ damage.

Background:

Line 38: Pregnancy-induced hypertension is a major cause of maternal and fetal morbidity and mortality.

Line 40: …hypertensive disorders exist, including…

Lines 43-53: I’m not sure if these definitions should be included as they make the background longer than it should be.

Line 54: Pre-eclampsia is defined as new-onset hypertension…

Line 55: …or ANY sign of end-organ damage…

Line 56: Please add – “Proteinuria, however, is not a requirement anymore to make a diagnosis of pre-eclampsia.” (Reference: PMID: 32063058)

Line 58: Disease severity varies, and it is considered severe if it is associated with one or more of the following: …

Line 59: … at least 6 hours apart, a 24-hour protein of >5 g, and/or other symptoms, such as…

Line 62: Please add – “The main underlying disease mechanism is endothelial dysfunction affecting multiple organs, including brain, liver, kidneys, and placenta.” (Reference: PMID: 32063058)

Line 66: Its global prevalence is extremely variable…

Line 69: In Africa, it occurs in 10% of pregnancies…

Line 70: Around 4 in 100 women experience PROBLEMS with high blood pressure and kidney dysfunction…

Line 75: …maternal deaths, most of which are…

Line 76: Liver and kidney dysfunction…common manifestations of end-organ damage…

Line 78: Hyperuricemia indicates kidney dysfunction because of decreased glomerular filtration, decreased tubular secretion, and/or increased proximal tubular reabsorption. (Reference: PMID: 15660336) It is also an independent risk factor for cardiovascular diseases as it has been suggested that it alters vascular function and mediates vascular inflammation. As such, hyperuricemia can perhaps predict the severity of pre-eclampsia.

Lines 81-85: Pre-eclampsia is also the most common cause of liver dysfunction in 3% of pregnancies because of microvesicular fat deposition and reduced blood flow to the liver potentially causing ischemia and periportal hemorrhage. (References: PMID 3189435 and PMID 8607495). ALT and AST are usually normal, but when they become elevated and are accompanied by abdominal pain, it almost always suggests the severe end of the disease spectrum.

Line 86: As such, this study was undertaken to evaluate the utility of levels of serum uric acid and liver function tests as early biomarkers of pre-eclampsia-related end organ damage…

Methods and Materials:

Line 91: An institution-based comparative cross-sectional study design…

Line 92: All patient had antenatal care follow-up…

Line 94: We excluded pregnant women who had…, those with a history of medication USE (eg…), and those with substance abuse (eg smoking and alcohol consumption) as these factors may affect the outcome variables.

Line 101: …to evaluate the levels of serum uric acid and liver function tests…

Line 104: …sample size calculations, like Minitab and Epi-info, …

Line 105: As an input, G-Power requires selecting appropriate test family (t-test in this case) and statistical test within test family (linear regression in this case) and specifying alpha error probability, …

Line 108: …(80%), and effect size…

Line 112: Trained nurses…

Line 113: Four laboratory technicians were oriented about blood sample collection and storage. Questionnaires were filled by face-to-face interviews with participants. Pre-pregnancy BMIs were calculated from…

Line 118: New sentence – “Two consecutive blood pressure readings were also measured.”

Line 122: It was then centrifuged…

Line 124: …refrigerator UNTIL…

Line 132: …to check IF there is any data inconsistency and to avoid any problem with data entry processes…

Line 138: All data were checked for cleaning and completeness. Data processing and analysis obtained from…

Line 144: …and other assumptions WERE checked before doing any statistical analysis and WERE fulfilled.

Line 147: WITH bivariable linear regression analysis, variables with…WHILE WITH multivariable linear regression analysis, variables with…

Line 152: Ethical clearance was obtained from…

Line 159: The present study recruited 102 female study participants, 51 of whom were pre-eclamptic pregnant women while 51 of whom were healthy pregnant women with gestational age above 20 weeks.

Line 161: The mean ages of pre-eclamptic patients and healthy controls were 32.9 years (range: 21-42) and 29.5 years (range: 23-35) RESPECTIVELY.

Line 162: The mean gestational ages of pre-eclamptic patients and healthy controls were 33.6 weeks (range: 22-39) and 29.3 weeks (range: 21-38) RESPECTIVELY.

Line 164: Majority of the study participants were married. I will delete the statement - “and the rest were single in both case and control groups.”

Line 166: Among the healthy controls, 41 (80.4%) were living in urban areas.

Line 176: Based on the calculation, …

Line 178: Majority of the healthy controls (94.1%) had normal BMI while the rest (5.9%) were overweight.

Line 179: None in the control group WAS underweight. Blood pressures of the study participants were recorded…

Line 186: I think it would be better to include the reference ranges for the different laboratory parameters in Table 3.

Line 188: Majority of the pre-eclamptic patients…but had normal total and direct bilirubin levels (60.8% and 62.7% respectively).

***Under paragraphs 3.3 and 3.4, the abbreviation SUA was used instead of serum uric acid. For consistency, the authors can start using the abbreviations at the start of the manuscript and make sure to spell them out on first usage and then use the abbreviations afterwards. Aside from SUA, the rule also applies to ALT and AST.

Lines 196-199: …statistically significant difference…

Line 200: There was, however, no significant difference in the total and direct bilirubin levels…

Line 202: The age and the gestational age of the study participants were also not significantly different…

Line 204: …compared to that of healthy controls…

Line 215: …; however, there was no significant difference in the mean SUA levels between the age groups 19-25 years and >35 years…

Line 219: …in mean ALT levels across different age groups…

Line 221: …; however, there was no significant difference in the mean ALT levels between the age groups…

***General comments for the results sections:

1. The word “level” should be inserted after all these phrases “mean SUA, ALT, and AST,” i.e. mean SUA level, mean ALT level, and mean AST level.

2. The preposition“of” can be deleted in all sentences containing the phrase “between the age groups,” i.e. between the age groups 19-25 years and 26-34 years. There is no need for the preposition “of” in this case.

3. To be grammatically correct, there should be a semicolon and a comma before and after the word “however” - …; however,…

Discussion:

Lines 303-306: I think that these statements need not be included in the discussion. I would consider starting the discussion by summarizing your key findings.

Line 307: In our study, majority of the pre-eclamptic patients were found…, which is in keeping with the results of a study conducted by…showing that advanced maternal age…

Line 309: Other studies conducted in Finland and Japan also demonstrated that…

Line 312: Our study also showed that the…were found to be…

Line 313: …, and there was no statistically significant difference between the two groups. This finding is in contrast in with the findings of other studies, and this difference may be due to…

Line 317: In our study, the mean serum uric acid level was significantly higher in cases compared to controls…, which is again consistent with the findings of other studies. I would elaborate more on these findings and explain the mechanism/s behind these findings. In addition, the mean serum ALT and AST levels were also significantly higher…

Line 323: …inflammatory response, resulting in endothelial dysfunction causing vasoconstriction and eventual liver and kidney dysfunction…

Lines 325-331: These statements are somewhat a repetition of the statements in lines 320-324. I would combine these 2 paragraphs to be more cohesive.

Line 332: On the other hand, our study showed that there was no significant difference in the total and direct bilirubin levels…

Line 334: Our findings are in contrast with the findings of other studies wherein there were significant increases in…

Line 336: …due to variations in the extent of…

Line 337: In pre-eclampsia, the main cause of hyperbilirubinemia is thought to be disseminated…

Line 340: In our study, there were significant differences in the mean SUA, ALT, and AST levels across the different age groups…The mean SUA level in the age group 26-34 years was 4.43 mg/dL, which then increased to 5.66 mg/dL in the age group >35 years. Please modify the sentences after this one to the same format.

Line 346: These findings were consistent with those of other studies, which showed that…

Line 348: Similarly, there were also significant differences in the mean SUA, ALT, and AST levels across the different gestational age groups…Please also modify the sentences after this one to the same format as mentioned above.

Line 357: Lastly, we performed a multivariable linear regression analysis and found a statistically significant positive association between…

Lines 362-365: If I am not mistaken, these statements are inaccurate as your study did not show any significant difference in the total and direct bilirubin levels between pre-eclamptic and healthy pregnant women.

Conclusion:

Our study revealed that serum uric acid, ALT, and AST levels were HIGHER in pre-eclamptic pregnant women compared to THOSE of healthy pregnant women, and the differences were statistically significant. As such, serum uric acid and liver function tests may be considered as early biomarkers of pre-eclampsia-related end organ damage.

6. PLOS authors have the option to publish the peer review history of their article (what does this mean?). If published, this will include your full peer review and any attached files.

Reviewer #1: No

Reviewer #2: No

---

## [Author Response · Author response to Decision Letter 0]

27 Apr 2022

April 27, 2022

Rebuttal letter

Pone-d-21-30928

Evaluation of serum uric acid and liver function tests among pregnant women with and without preeclampsia at the University of Gondar comprehensive specialized hospital, northwest Ethiopia

Dear Editors and Reviewers,

We appreciate the constructive comments and suggestions for improving our manuscript. Based on the comments and suggestions, we have made corrections and modifications and provided point-by-point responses to the comments and suggestions. Please find our responses in green under the comments and suggestions presented by academic editors and reviewers in yellow. Below are our responses to each point raised by the academic editor and reviewers.

Best regards!

Tadesse Asmamaw Dejenie, on behave of all authors

Response to academic editor’s comment

Comment 1: Please ensure that your manuscript meets PLOS ONE's style requirements, including those for file naming

Author response: Thank you very much for sending the link to the PLOS ONE style templates; we have taken your advice and updated our manuscript to meet the style requirements of PLOS ONE. Please check it in the revised manuscript.

Comment 2a: Please clarify the sources of funding (financial or material support) for your study. List the grants or organizations that supported your study, including funding received from your institution. 

Author response: We'd like to thank you for your insightful remarks and suggestions. We've considered the suggestions and made the necessary revisions. For this work, we did not receive any funds.

Comment 2b: State what role the funders took in the study. If the funders had no role in your study, please state: “The funders had no role in study design, data collection and analysis, decision to publish, or preparation of the manuscript.”

Author response: We stated that we did not receive any funding for this study and that the University of Gondar was the study setting.

Comment 2c: If any authors received a salary from any of your funders, please state which authors and which funders.

Author response: No author received a salary since there is no funder for this work.

Comment 2d: If you did not receive any funding for this study, please state: “The authors received no specific funding for this work.

Author response: Many thanks for your suggestion; in the revised manuscript, we stated that "the authors received no specific funding for this work." Please check it in the revised manuscript. 

Comment 3: We note that you have provided additional information within the Acknowledgements Section that is not currently declared in your Funding Statement. Please note that funding information should not appear in the Acknowledgments section or other areas of your manuscript. We will only publish funding information present in the Funding Statement section of the online submission form. 

Author response: We'd like to thank you for your insight. This section has been removed entirely. For further information, kindly check the revised manuscript.

Comment 4: Please complete your Competing Interests on the online submission form to state any Competing Interests. If you have no competing interests, please state "The authors have declared that no competing interests exist."

Author response: We are grateful for your suggestions and stated that the authors have declared that no competing interests exist." Please find it in the revised manuscript. 

Comment 5: In your Data Availability statement, you have not specified where the minimal data set underlying the results described in your manuscript can be found. Upon re-submitting your revised manuscript, please upload your study’s minimal underlying data set as either Supporting Information files or to a stable, public repository and include the relevant URLs, DOIs, or accession numbers within your revised cover letter. 

Author response: We'd want to express our gratitude once more for bringing this to our attention. We stated that the authors confirm that all data underlying the findings are fully available without restriction. All relevant data are within the manuscript in the revised manuscript. Please check it. 

Response to Reviewers’ comment

Reviewer #1: 

Minor comments: 

Comment 1: Table 4: it is not clear which column is "Cases" and which is "Controls"

Author response: Thank you so much; we've made the necessary modifications. Please check it in the revised manuscript.

Comment 2: t-test was applied on several spots where it was obvious from presented data that CV was above 30%

Author response: Thank you very much for taking the time to share your insights. We used an independent t-test because we thought it would be a reasonable statistical model test for comparing measurable parameters across case and control groups in the case of this study.

Comment 3: The authors did not address early vs. late preeclampsia differences.

Author response: We really appreciate your insights. We considered that it is obvious that the complication is more common in late preeclamptic women than in early preeclamptic women.

Comment 4: Comparison studies of biochemical markers across gestational age and different age groups would look better it were presented in tables rather than textually.

Author response: We sincerely appreciate your insight. Since we were afraid of having a bulky table in our manuscript, we removed those suggested tables, but we have now inserted these tables as per your suggestion. Please find tables 5 and 6 of the revised manuscript. 

Major comments: 

Comment 1: Based on the results we can only conclude that there are differences among PE vs normotensive pregnancies. However the authors make a conclusion of SUA, ALT and AST as early biomarkers of preeclampsia. This is incorrect, as they did not report when the samples were collected. Based on their study design, these women have already been diagnosed with PE so increase of these biochemical parameters could be the consequence of the disease, not early marker of it.

Author response: Thank you so much for everything. We've taken your concern into account and made the necessary modifications. Please check it in the revised manuscript. 

Comment 2: The authors report that women with preeclampsia were older compared to the control group. In the same time they observe positive association between age and biochemical markers such as ALT and AST. It is possible that age is actually pulling the difference so authors should have performed adequate statistical analysis to check this.

Author response: Thanks, Even though we have stated that women with preeclampsia are older than the control groups, their age difference is insignificant. Please see table 4 of the manuscript. 

Comment 3: I am confused that the authors did not observe the difference in bilirubin concentrations between cases and controls, but report "being preeclampsia shows significant association with total bilirubin in multiple linear regressions"

Author response: Please accept our apology for the confusion; we have already done that in table 4 of the manuscript. I think this is because of Table 4; it was not clear which column was "cases" and which was "controls" for you. Now we separate the case and control columns in this table. Please find it in the revised manuscript.

Reviewer #2: 

Abstract:

Suggestion 1: Line 11: Pre-eclampsia can be described as new-onset hypertension…

Author response: Indeed, thank you very much for your suggestion to improve the clarity of our manuscript. We have modified it as per your suggestion. Please check it in line 11 of the revised manuscript.

Suggestion 2: Line 12: …or ANY indication…

Author response: Indeed, thank you very much for your suggestion to improve the clarity of our manuscript. We have modified it as per your suggestion. Please check it in line 12 of the revised manuscript.

Suggestion 3: Line 13: Liver and kidney dysfunction…

Author response: Indeed, thank you very much for your suggestion to improve the clarity of our manuscript. We have modified it as per your suggestion. Please check it in line 13 of the revised manuscript.

Suggestion 4: Line 14: …common MANIFESTATIONS of end-organ damage…

Author response: Indeed, thank you very much for your suggestion to improve the clarity of our manuscript. We have modified it as per your suggestion. Please check it in line 14 of the revised manuscript.

Suggestion 5: Line 15: …most common cause of LIVER and KIDNEY dysfunction… (Please change all “renal” terms to “kidney” instead.)

Author response: Indeed, thank you very much for your suggestion to improve the clarity of our manuscript. We have modified it as per your suggestion. Please check it in line 15 & 16 of abstract section of the manuscript and throughout the revised manuscript.

Suggestion 6: Line 15: Pre-eclampsia is the most common cause of liver and kidney dysfunction – in what patient population? – Due to hypoxia and endothelial dysfunction.

Author response: In a population of pregnant women, ischemia caused by hypoperfusion of the placenta leads to the release of many mediators, causing endothelial dysfunction and a cascade of systemic disorders including hypertension, proteinuria, edema, and platelet aggregation (Szpera-Gozdziewicz et al., 2014).

Suggestion 7: Line 17: Hyperuricemia indicates kidney dysfunction…

Author response: Author response: Indeed, thank you very much for your suggestion to improve the clarity of our manuscript. We have modified it as per your suggestion. Please check it in line 17 of the revised manuscript.

Suggestion 8: Line 18: …severity of pre-eclampsia; therefore, the objective of this study is to evaluate the utility of the levels of serum uric acid and liver function tests [alanine aminotransferase (ALT) and aspartate aminotransferase (AST)] as early biomarkers…

Author response: Indeed, thank you very much for your suggestion to improve the clarity of our manuscript. We have modified it as per your suggestion. Please check it in lines 18-20 of the revised manuscript.

Suggestion 9: Line 20: An institution-based comparative cross-sectional study design…

Author response: Author response: Indeed, thank you very much for your suggestion to improve the clarity of our manuscript. We have modified it as per your suggestion. Please check it in line 21 of the revised manuscript.

Suggestion 10: Line 21: …conducted,

Author response: Author response: Indeed, thank you very much for your suggestion to improve the clarity of our manuscript. We have modified it as per your suggestion. Please check it in line 21 of the revised manuscript.

Suggestion 11: Line 22: The parameters measured were levels of serum uric acid and liver function tests.

Author response: Indeed, thank you very much for your suggestion to improve the clarity of our manuscript. We have modified it as per your suggestion. Please check it in line 23 of the revised manuscript.

Suggestion 12: Line 24: There were statistically significant differences in the mean serum uric acid, ALT, and AST levels …

Author response: Indeed, thank you very much for your suggestion to improve the clarity of our manuscript. We have modified it as per your suggestion. Please check it in line 24-26 of the revised manuscript.

Suggestion 13: Line 26: There were no statistically significant differences in the mean total and direct bilirubin levels…

Author response: Indeed, thank you very much for your suggestion to improve the clarity of our manuscript. We have modified it as per your suggestion. Please check it in line 27 of the revised manuscript.

Suggestion 14: Line 27: There were also statistically significant differences in the mean serum uric acid, ALT, and AST levels across different gestational age categories.

Author response: Indeed, thank you very much for your suggestion to improve the clarity of our manuscript. We have modified it as per your suggestion. Please check it in lines 27 & 28 of the revised manuscript.

Suggestion 15: Line 30: Our study revealed that serum uric acid, ALT, and AST levels were HIGHER in pre-eclamptic pregnant women compared to THOSE of healthy pregnant women, and the differences were statistically significant. As such, serum uric acid and liver function tests may be considered as early biomarkers of pre-eclampsia-related end organ damage.

Author response: Indeed, thank you very much for your suggestion to improve the clarity of our manuscript. We have modified the conclusion section of this manuscript as per your and reviewer #1 suggestions. Please check it in the conclusion and abstraction section of the revised manuscript.

Introduction:

Suggestion 16: Line 38: Pregnancy-induced hypertension is a major cause of maternal and fetal morbidity and mortality.

Author response: Indeed, thank you very much for your suggestion to improve the clarity of our manuscript. We have modified it as per your suggestion. Please check it in lines 36 & 37 of the revised manuscript.

Suggestion 17: Line 40: …hypertensive disorders exist, including…

Author response: Indeed, thank you very much for your suggestion to improve the clarity of our manuscript. We have modified it as per your suggestion. Please check it in line 37 of the revised manuscript.

Suggestion 18: Lines 43-53: I’m not sure if these definitions should be included as they make the background longer than it should be.

Author response: Indeed, thank you very much for your suggestion to improve the clarity of our manuscript. We have accepted and deleted it as per your suggestion. Please check it in the revised manuscript.

Suggestion 19: Line 54: Pre-eclampsia is defined as new-onset hypertension…

Author response: Indeed, thank you very much for your suggestion to improve the clarity of our manuscript. We have modified it as per your suggestion. Please check it in line 45 of the revised manuscript.

Suggestion 20: Line 55: …or ANY sign of end-organ damage…

Author response: Indeed, thank you very much for your suggestion to improve the clarity of our manuscript. We have modified it as per your suggestion. Please check it in line 46 of the revised manuscript.

Suggestion 21: Line 56: Please add – “Proteinuria, however, is not a requirement anymore to make a diagnosis of pre-eclampsia.” (Reference: PMID: 32063058)

Author response: Thank you very much for your suggestion to improve the clarity of our manuscript. We have added and cited the above-mentioned sentences as per your suggestion. Please check it in lines 48-49 of the revised manuscript.

Suggestion 22: Line 58: Disease severity varies, and it is considered severe if it is associated with one or more of the following: …

Author response: Thank you so much for your suggestion to improve the clarity of our manuscript. We have modified it as per your suggestion. Please check it in line 50 of the revised manuscript

Suggestion 23: Line 59: … at least 6 hours apart, a 24-hour protein of >5g, and/or other symptoms, such as…

Author response: Indeed, thank you very much for your suggestion to improve the clarity of our manuscript. We have modified it as per your suggestion. Please check it in line 52 of the revised manuscript.

Suggestion 24: Line 62: Please add – “The main underlying disease mechanism is endothelial dysfunction affecting multiple organs, including brain, liver, kidneys, and placenta.” (Reference: PMID: 32063058)

Author response: Thank you very much for your suggestion to improve the clarity of our manuscript. We have added and cited the above-mentioned sentences as per your suggestion. Please check it in lines 53-54 of the revised manuscript.

Suggestion 25: Line 66: Its global prevalence is extremely variable…

Author response: Thank you so much for your advice on how to make our manuscript clear. We've made the changes you suggested. Please double-check it in the revised manuscript on line 61.

Suggestion 26: Line 69: In Africa, it occurs in 10% of pregnancies…

Author response: Thank you so much for your suggestions on how to improve the clarity of our manuscript. We've made the modifications you recommended. Please check it on lines 63-64 of the revised manuscript.

Suggestion 27: Line 70: Around 4 in 100 women experience PROBLEMS with high blood pressure and kidney dysfunction…

Author response: Thank you for your suggestions for improving the clarity of our manuscript. Your suggestions have been accepted. In the revised manuscript, on lines 64-65, please double-check it.

Suggestion 28: Line 75: …maternal deaths, most of which are…

Indeed, thank you very much for your suggestion to improve the clarity of our manuscript. We have modified it as per your suggestion. Please check it in lines 68-69 of the revised manuscript

Suggestion 29: Line 76: Liver and kidney dysfunction…common manifestations of end-organ damage…

Author response: Thank you for your suggestions for improving the clarity of our manuscript. Your suggestions have been accepted. Please double-check lines 70-71 in the revised manuscript.

Suggestion 30: Line 78: Hyperuricemia indicates kidney dysfunction because of decreased glomerular filtration, decreased tubular secretion, and/or increased proximal tubular reabsorption. (Reference: PMID: 15660336) It is also an independent risk factor for cardiovascular diseases as it has been suggested that it alters vascular function and mediates vascular inflammation. As such, hyperuricemia can perhaps predict the severity of pre-eclampsia.

Author response: Thank you very much for your suggestion to improve the clarity of our manuscript. We have added and cited the above-mentioned sentences as per your suggestion. Please check it in lines 72-73 of the revised manuscript.

Suggestion 31: Lines 81-85: Pre-eclampsia is also the most common cause of liver dysfunction in 3% of pregnancies because of micro vesicular fat deposition and reduced blood flow to the liver potentially causing ischemia and periportal hemorrhage. (References: PMID 3189435 and PMID 8607495). ALT and AST are usually normal, but when they become elevated and are accompanied by abdominal pain, it almost always suggests the severe end of the disease spectrum. 

Author response: Thank you very much for your suggestion to improve the clarity of our manuscript. We have added and cited the above-mentioned sentences as per your suggestion. Please check it in lines 77-82 of the revised manuscript

Suggestion 32: Line 86: As such, this study was undertaken to evaluate the utility of levels of serum uric acid and liver function tests as early biomarkers of pre-eclampsia-related end organ damage…

Author response: Thank you for your suggestions on making our manuscript more clear. Your recommendations have been taken into consideration. Please double-check lines 83–85 in the revised manuscript.

Methods and Materials:

Suggestion 33: Line 91: An institution-based comparative cross-sectional study design…

Author response: Thank you for your suggestions on making our manuscript more clear. Your recommendation has been accepted, and we have modified it. Please see line 88 of the revised manuscript.

Suggestion 34: Line 92: All patients had antenatal care follow-up…

Author response: Thank you for your suggestions on making our manuscript more clear. Your recommendation has been accepted, and we have modified it. Please see lines 89-90 of the revised manuscript.

Suggestion 35: Line 94: We excluded pregnant women who had…, those with a history of medication USE (eg…), and those with substance abuse (eg smoking and alcohol consumption) as these factors may affect the outcome variables.

Author response: you for your suggestions on making our manuscript more clear. Your recommendation has been accepted, and we have modified it. Please see lines 91-94 of the revised manuscript.

Suggestion 36: Line 101: …to evaluate the levels of serum uric acid and liver function tests…

Author response: Thank you for your suggestions on making our manuscript more clear. Your recommendation has been accepted, and we have modified it. Please see line 88 of the revised manuscript.

Suggestion 37: Line 104: …sample size calculations, like Minitab and Epi-info, …

Author response: Thank you for your suggestions on making our manuscript more clear. Your recommendation has been accepted, and we have modified it. Please see line 101 of the revised manuscript.

Suggestion 38: Line 105: As an input, G-Power requires selecting appropriate test family (t-test in this case) and statistical test within test family (linear regression in this case) and specifying alpha error probability, …

Author response: Thank you for your suggestions on making our manuscript more clear. Your recommendation has been accepted, and we have modified it. Please see lines 102-104 of the revised manuscript.

Suggestion 39: Line 108: …(80%), and effect size…

Author response: Thank you for your suggestions on making our manuscript more clear. We've taken your suggestion and made modifications to it. Please check line 105 of the revised manuscript.

Suggestion 40: Line 112: Trained nurses…

Author response: Thank you for your suggestions on making our manuscript more clear. We've taken your suggestion and made modifications to it. Please check line 109 of the revised manuscript.

Suggestion 41: Line 113: Four laboratory technicians were oriented on blood sample collection and storage. Questionnaires were filled out through face-to-face interviews with participants. Pre-pregnancy BMIs were calculated from…

Author response: Thank you for your suggestions on making our manuscript more clear. We've taken your suggestion and made modifications to it. Please check lines 110-112 of the revised manuscript.

Suggestion 42: Line 118: New sentence – “Two consecutive blood pressure readings were also measured.”

Author response: Thank you for your suggestions on making our manuscript more clear. We've added this new sentence to our manuscript to make it more clear. Please line 116 of the revised manuscript.

Suggestion 43: Line 122: It was then centrifuged…

Author response: Thank you for your suggestions on making our manuscript more clear. We've taken your suggestion and made modifications to it. Please check line 19 of the revised manuscript.

Suggestion 44: Line 124: …refrigerator UNTIL…

Author response: Thank you for your suggestions on improving the clarity of our manuscript. We've taken your advice and modified it. Please check line 121 of the revised manuscript.

Suggestion 45: Line 132: …to check IF there is any data inconsistency and to avoid any problem with data entry processes…

Author response: Thank you for your suggestions on improving the clarity of our manuscript. We've taken your advice and modified it. Please check line 129 of the revised manuscript.

Suggestion 46: Line 138: All data were checked for cleaning and completeness. Data processing and analysis obtained from…

Author response: Thank you for your suggestions on improving the clarity of our manuscript. We've taken your advice and modified it. Please check line 135 of the revised manuscript.

Suggestion 47: Line 144: …and other assumptions WERE checked before doing any statistical analysis and WERE fulfilled. 

Author response: Thank you for your suggestions on improving the clarity of our manuscript. We've taken your advice and modified it. Please check lines 141 & 142 of the revised manuscript.

Suggestion 48: Line 147: WITH bivariable linear regression analysis, variables with…WHILE WITH multivariable linear regression analysis, variables with…

Author response: Thank you for your suggestions on improving the clarity of our manuscript. We've taken your advice and modified it. Please check lines 145 & 146 of the revised manuscript.

Suggestion 49: Line 152: Ethical clearance was obtained from…

Author response: Thank you for your suggestions on improving the clarity of our manuscript. We've taken your advice and modified it. Please check line 402 of the revised manuscript.

Suggestion 50: Line 159: The present study recruited 102 female study participants, 51 of whom were pre-eclamptic pregnant women while 51 of whom were healthy pregnant women with gestational age above 20 weeks.

Author response: Thank you for your suggestions on improving the clarity of our manuscript. We've taken your advice and modified it. Please check lines 150-152 of the revised manuscript.

Suggestion 51: Line 161: The mean ages of pre-eclamptic patients and healthy controls were 32.9 years (range: 21-42) and 29.5 years (range: 23-35) RESPECTIVELY. Line 162: The mean gestational ages of pre-eclamptic patients and healthy controls were 33.6 weeks (range: 22-39) and 29.3 weeks (range: 21-38) RESPECTIVELY.

Author response: Thank you for your suggestions on improving the clarity of our manuscript. We've taken your advice and modified it. Please check lines 155-159 of the revised manuscript.

Suggestion 52: Line 164: Majority of the study participants were married. I will delete the statement - “and the rest were single in both case and control groups.”

Author response: Thank you for your suggestions on improving the clarity of our manuscript. We've accepted and modified it. Please check line 155 of the revised manuscript.

Suggestion 53: Line 166: Among the healthy controls, 41 (80.4%) were living in urban areas.

Author response: Thank you for your suggestions on improving the clarity of our manuscript. We've taken your advice and modified it. Please check lines 156-159 of the revised manuscript.

Suggestion 54: Line 176: Based on the calculation, …

Author response: Thank you for your suggestions on improving the clarity of our manuscript. We've taken your advice and modified it. Please check line 171 of the revised manuscript.

Suggestion 55: Line 178: Majority of the healthy controls (94.1%) had normal BMI while the rest (5.9%) were overweight. None in the control group WAS underweight. Blood pressures of the study participants were recorded…

Author response: Thank you for your suggestions on improving the clarity of our manuscript. We've taken your advice and modified it. Please check lines 173-175 of the revised manuscript.

Suggestion 56: Line 186: I think it would be better to include the reference ranges for the different laboratory parameters in Table 3.

Author response: Thanks to your suggestion, we've included the reference ranges for the different laboratory parameters in Table 3. Please check in table 3 of the revised manuscript.

Suggestion 57: Line 188: Majority of the pre-eclamptic patients…but had normal total and direct bilirubin levels (60.8% and 62.7% respectively).

Author response: Thank you for your suggestions on improving the clarity of our manuscript. We've taken your advice and modified it. Please check lines 183-185 of the revised manuscript.

Suggestion 58: ***Under paragraphs 3.3 and 3.4, the abbreviation SUA was used instead of serum uric acid. For consistency, the authors can start using the abbreviations at the start of the manuscript and make sure to spell them out on first usage and then use the abbreviations afterwards. Aside from SUA, the rule also applies to ALT and AST.

Author response: Thank you for your suggestions. We've corrected it. Please check it throughout the revised manuscript.

Suggestion 59: Lines 196-199: …statistically significant difference…

Author response: Thank you for your suggestions on improving the clarity of our manuscript. We've taken your advice and modified it. Please check lines 190-193 of the revised manuscript.

Suggestion 60: Line 200: There was, however, no significant difference in the total and direct bilirubin levels…

Author response: Thank you for your suggestions on improving the clarity of our manuscript. We've taken your advice and modified it. Please check line 194 of the revised manuscript.

Suggestion 61: Line 202: The age and the gestational age of the study participants were also not significantly different…

Author response: Thank you for your suggestions on improving the clarity of our manuscript. We've taken your advice and modified it. Please check line 197 of the revised manuscript.

Suggestion 62: Line 204: …compared to that of healthy controls…

Author response: Thank you for your suggestions on improving the clarity of our manuscript. We've taken your advice and modified it. Please check line 199 of the revised manuscript.

Suggestion 63: Line 215: …; however, there was no significant difference in the mean SUA levels between the age groups 19-25 years and >35 years…

Author response: Thank you for your suggestions on making our manuscript more clear. We've taken your suggestion and modified it. Kindly check line 208 of the revised manuscript.

Suggestion 64: Line 219: …in mean ALT levels across different age groups…

Author response: Thank you for your suggestions on making our manuscript more clear. We've taken your suggestion and modified it. Kindly check lines 213-215 of the revised manuscript.

Suggestion 65: Line 221: …; however, there was no significant difference in the mean ALT levels between the age groups…

Author response: Thank you for your suggestions on making our manuscript more clear. We've taken your suggestion and modified it. Kindly check lines 213-215 of the revised manuscript.

Suggestion 66: ***General comments for the results sections: 

1. The word “level” should be inserted after all these phrases “mean SUA, ALT, and AST,” i.e. mean SUA level, mean ALT level, and mean AST level.

2. The preposition“of” can be deleted in all sentences containing the phrase “between the age groups,” i.e. between the age groups 19-25 years and 26-34 years. There is no need for the preposition “of” in this case.

3. To be grammatically correct, there should be a semicolon and a comma before and after the word “however” - …; however,…

Author response: Thank you for your suggestions on how to improve the clarity of our manuscript. We've made significant changes based on your suggestions. Please check throughout the result section of our revised manuscript. 

Discussion:

Suggestion 67: Lines 303-306: I think that these statements need not be included in the discussion. I would consider starting the discussion by summarizing your key findings.

Author response: As per your suggestions, we have deleted and made significant changes. Please find it in the first paragraph of the discussion section of the revised manuscript. 

Suggestion 68: Line 307: In our study, majority of the pre-eclamptic patients were found…, which is in keeping with the results of a study conducted by…showing that advanced maternal age…

Author response: Thank you for your suggestions on improving the clarity of our manuscript. We took your suggestion and modified it. Please take a look at lines 306-308 of the revised manuscript.

Suggestion 69: Line 309: Other studies conducted in Finland and Japan also demonstrated that…

Author response: Thank you for your suggestions on making our manuscript more clear. We acknowledged your suggestion and modified it to make it better. Please see line 309 of the revised manuscript for more information.

Suggestion 70: Line 312: Our study also showed that the…were found to be…

Author response: Thank you for your suggestions on making our manuscript more clear. We acknowledged your suggestion and modified it to make it better. Please check it line 312 of the revised manuscript for more information.

Suggestion 72: Line 313: …, and there was no statistically significant difference between the two groups. This finding is in contrast in with the findings of other studies, and this difference may be due to…

Author response: Thank you for your suggestions on improving the clarity of our manuscript. We accepted your suggestion into consideration and improved it. More information can be found on lines 313-115 of the revised manuscript.

Suggestion 73: Line 317: In our study, the mean serum uric acid level was significantly higher in cases compared to controls…, which is again consistent with the findings of other studies. I would elaborate more on these findings and explain the mechanism/s behind these findings. In addition, the mean serum ALT and AST levels were also significantly higher…

Author response: Thank you for your suggestions. We've corrected it. Please check lines 317–319 of the revised manuscript.

Suggestion 74: Line 323: …inflammatory response, resulting in endothelial dysfunction causing vasoconstriction and eventual liver and kidney dysfunction…

Author response: Thank you so much for your suggestions. It has now been addressed. Please check lines 321–322 of the revised manuscript.

Suggestion 75: Lines 325-331: These statements are somewhat a repetition of the statements in lines 320-324. I would combine these 2 paragraphs to be more cohesive.

Author response: Thank you so much for your advice. We gratefully accepted and modified the recommendation. Please check the revised manuscript.

Suggestion 76: Line 332: On the other hand, our study showed that there was no significant difference in the total and direct bilirubin levels…

Author response: Thank you for your suggestions on making our manuscript more clear. Your recommendation has been accepted, and we have modified it. Please see line 325 of the revised manuscript.

Suggestion 77: Line 334: Our findings are in contrast with the findings of other studies wherein there were significant increases in…

Author response: Thank you for your suggestions on making our manuscript more clear. Your recommendation has been accepted, and we have modified it. Please see line 327 of the revised manuscript.

Suggestion 78: Line 336: …due to variations in the extent of…

Author response: Thank you for your suggestions on making our manuscript more clear. Your recommendation has been accepted, and we have modified it. Please see line 330 of the revised manuscript.

Suggestion 79: Line 337: In pre-eclampsia, the main cause of hyperbilirubinemia is thought to be disseminated…

Author response: Thank you for your suggestions on making our manuscript more clear. Your recommendation has been accepted, and we have modified it. Please see line 331 of the revised manuscript.

Suggestion 80: Line 340: In our study, there were significant differences in the mean SUA, ALT, and AST levels across the different age groups…The mean SUA level in the age group 26-34 years was 4.43 mg/dL, which then increased to 5.66 mg/dL in the age group >35 years. Please modify the sentences after this one to the same format.

Author response: Thank you for your suggestions on making our manuscript more clear. Your recommendation has been accepted, and we have modified it. Please see lines 333-336 of the revised manuscript.

Suggestion 81: Line 346: These findings were consistent with those of other studies, which showed that…

Author response: Thank you for your suggestions on making our manuscript more clear. Your recommendation has been accepted, and we have modified it. Please see line 339 of the revised manuscript.

Suggestion 82: Line 348: Similarly, there were also significant differences in the mean SUA, ALT, and AST levels across the different gestational age groups…Please also modify the sentences after this one to the same format as mentioned above.

Author response: Thank you for your suggestions on improving the clarity of our manuscript. Your suggestion has been accepted, and it has been modified. Please find lines 341-342 of the revised manuscript.

Suggestion 83: Line 357: Lastly, we performed a multivariable linear regression analysis and found a statistically significant positive association between…

Author response: Thank you for your suggestions on improving the clarity of our manuscript. Your suggestion has been accepted, and it has been modified. Please find line 350 of the revised manuscript.

Suggestion 84: Lines 362-365: If I am not mistaken, these statements are inaccurate as your study did not show any significant difference in the total and direct bilirubin levels between pre-eclamptic and healthy pregnant women.

Author response: We appreciate your insight. We accepted it and made significant changes.

Conclusion:

Suggestion 80: Our study revealed that serum uric acid, ALT, and AST levels were HIGHER in pre-eclamptic pregnant women compared to THOSE of healthy pregnant women, and the differences were statistically significant. As such, serum uric acid and liver function tests may be considered as early biomarkers of pre-eclampsia-related end organ damage.

Author response: Indeed, thank you very much for your suggestion to improve the clarity of our manuscript. We have modified the conclusion section of the revised manuscript as per your and other reviewers' suggestions. Please check the revised manuscript.

---

## [Editor Report · Decision Letter 1]

24 Jun 2022

PONE-D-21-30928R1EVALUATION OF SERUM URIC ACID AND LIVER FUNCTION TESTS AMONG PREGNANT WOMEN WITH AND WITHOUT PREECLAMPSIA AT UNIVERSITY OF GONDAR COMPREHENSIVE SPECIALIZED HOSPITAL, NORTHWEST ETHIOPIAPLOS ONE

Dear Dr. Dejenie,

Thank you for submitting your manuscript to PLOS ONE. After careful consideration, we feel that it has merit but does not fully meet PLOS ONE’s publication criteria as it currently stands. Therefore, we invite you to submit a revised version of the manuscript that addresses the points raised during the review process.

Thank you for addressing reviewers' comments on the first revision of your manuscript.

Please address the following:

1. Describe the definition that was used to diagnose preeclampsia in this study, were the controls having normal BP? if so please refer to controls as normotensive pregnancy.

2. Combine the information in table 1 and table 2 into one larger table with "baseline characteristics". Consider adding p value

3. Please change  the column names in tables 1, 2 and 4 from "Case" and "Control" to "Preeclampsia" and "Normotensive Pregnancy"4. In table 2, rather than using the words: "below normal", "normal", "above normal"; provide values of BMI, SBP, DBP. Also add proteinuria values for "cases" and "Controls", add platelet cell counts for both groups

5. Consider deleting table 3, it does not provide extra information and becomes redundant with the information provided in table 4

6. Please review the format of the columns and the name of the columns of tables 5 and 6; it is not clear what it is being compared, and it is important for these tables to be as clear as possible, since it is the main evidence of your discussion

7. In table 7, consider deleting the following rows: marital status, religion, education, occupation and residence. They should not be important in order to address the difference in the lab values you are measuring.

8. Please review your conclusion, while I agree that your "study revealed that serum uric acid, ALT, and AST levels were higher in pre-eclamptic pregnant women compared to those of healthy pregnant women, and the differences were

statistically significant."

I disagree that "serum uric acid and liver function tests may be considered as biomarkers of pre-eclampsia-related end-organ damage." Please review the wording used, as your manuscript does not support the use of these values as biomarkers, rather it supports that these values are different, but not to be used on their own for the diagnosis of end organ damage of preeclampsia.

9. Please clarify if any of the patients with preeclampsia in this study were diagnosed with HELLP syndrome.

10. Please provide information on the maternal and fetal outcomes of those with preeclampsia in this study when compared to normal pregnancy, including maternal and fetal survival, preterm delivery, birth weight, etc.

We look forward to receiving your revised manuscript.

Kind regards,

Maria Lourdes Gonzalez Suarez, MD, PhD

Academic Editor

PLOS ONE

---

## [Author Response · Author response to Decision Letter 1]

1 Jul 2022

June 30, 2022

Rebuttal letter

Pone-d-21-30928

Evaluation of serum uric acid and liver function tests among pregnant women with and without preeclampsia at the University of Gondar comprehensive specialized hospital, northwest Ethiopia

Dear Editors and Reviewers,

We appreciate the constructive comments and suggestions for improving our manuscript. Based on the comments and suggestions, we have made corrections and modifications and provided point-by-point responses to the comments and suggestions. Please find our responses in green under the comments and suggestions presented by the academic editors in yellow. Below are our responses to each point raised by the academic editor.

Best regards!

Tadesse Asmamaw Dejenie, on behave of all authors

Response to academic editor’s comment

Comment 1: Describe the definition that was used to diagnose preeclampsia in this study, were the controls having normal BP? If so please refer to controls as normotensive pregnancy.

Author response: We'd like to thank you for your insightful remarks and suggestions. We have selected those preeclampsia women confirmed by physicians (high blood pressure ≥ 140/90 mmHg after 20 weeks of pregnancy and having proteinuria)

Comment 2: Combine the information in table 1 and table 2 into one larger table with "baseline characteristics". Consider adding p value

Author response: Indeed, thank you very much for your suggestion to improve the clarity of our manuscript. We have combined table1 and table 2 as per your suggestion. Please check it in the revised manuscript.

Comment 3: Please change the column names in tables 1, 2 and 4 from "Case" and "Control" to "Preeclampsia" and "Normotensive Pregnancy"

Author response: Thank you so much; we've accepted your suggestions and made the necessary modifications. Please check it in the revised manuscript.

Comment 4: In table 2, rather than using the words: "below normal", "normal", "above normal"; provide values of BMI, SBP, and DBP. Also add proteinuria values for "cases" and "Controls", add platelet cell counts for both groups

Author response: Indeed, thank you very much for your suggestion to improve the clarity of our manuscript. We have modified the words: "below normal", "normal", and "above normal". We have provided values of BMI, SBP, and DBP as per your suggestion but we did not do proteinuria and platelet cell count for this study. Please find it in table 1 of the revised manuscript.

Comment 5: Consider deleting table 3, it does not provide extra information and becomes redundant with the information provided in table 4

Author response: We want to express our gratitude for your expertise. This table has been deleted. For more information, kindly check out the revised manuscript

Comment 6: Please review the format of the columns and the name of the columns of tables 5 and 6; it is not clear what it is being compared, and it is important for these tables to be as clear as possible, since it is the main evidence of your discussion. 

Author response: We sincerely appreciate your insight. We removed those tables (5 and 6) since the information is already described textually. We did a post hoc (Bonferroni) test to see the SUA, ALT, and AST levels across different age and gestational age of study participants.

Comment 7: In table 7, consider deleting the following rows: marital status, religion, education, occupation and residence. They should not be important in order to address the difference in the lab values you are measuring. 

Author response: Indeed, thank you very much for your suggestion to improve the clarity of our manuscript. We have accepted your suggestion and deleted the rows as per your suggestion. Please check it in the revised manuscript.

Comment 8: Please review your conclusion, while I agree that your "study revealed that serum uric acid, ALT, and AST levels were higher in pre-eclamptic pregnant women compared to those of healthy pregnant women, and the differences were

statistically significant."

Author response: Thanks, We've taken your concern into account and made the necessary modifications. Please check the conclusion section of the revised manuscript.

Comment 9: Please clarify if any of the patients with preeclampsia in this study were diagnosed with HELLP syndrome

Author response: Thank you so much for your insight. During the study period, none of the pregnant women were diagnosed with HELLP syndrome.

Comment 10: Please provide information on the maternal and fetal outcomes of those with preeclampsia in this study when compared to normal pregnancy, including maternal and fetal survival, preterm delivery, birth weight, etc.

Author response: We really appreciate your thoughtful suggestions. A longitudinal study is required to provide information on maternal and fetal outcomes such as maternal and fetal survival, preterm delivery, and birth weight, but this is a cross-sectional study. At the time of data collection, those normotensive pregnant women came to the hospital for antenatal care follow up and those preeclampsia women were admitted to the University of Gondar hospital’s labor ward for the treatment of preeclampsia.

---

## [Editor Report · Decision Letter 2]

14 Jul 2022

EVALUATION OF SERUM URIC ACID AND LIVER FUNCTION TESTS AMONG PREGNANT WOMEN WITH AND WITHOUT PREECLAMPSIA AT UNIVERSITY OF GONDAR COMPREHENSIVE SPECIALIZED HOSPITAL, NORTHWEST ETHIOPIA

PONE-D-21-30928R2

Dear Dr. Dejenie,

We’re pleased to inform you that your manuscript has been judged scientifically suitable for publication and will be formally accepted for publication once it meets all outstanding technical requirements.

Kind regards,

Maria Lourdes Gonzalez Suarez, MD, PhD

Academic Editor

PLOS ONE

Additional Editor Comments (optional):

Thank you for addressing our comments and concerns.
---

## [Editor Report · Acceptance letter]

21 Jul 2022

PONE-D-21-30928R2 

 Evaluation of serum uric acid and liver function tests among pregnant women with and without preeclampsia at the University of Gondar comprehensive specialized hospital, Northwest Ethiopia 

Dear Dr. Dejenie:

I'm pleased to inform you that your manuscript has been deemed suitable for publication in PLOS ONE. Congratulations! Your manuscript is now with our production department. 

Kind regards, 

on behalf of

Dr. Maria Lourdes Gonzalez Suarez 

Academic Editor

PLOS ONE